# A Framework of Faster CRNN and VGG16-Enhanced Region Proposal Network for Detection and Grade Classification of Knee RA

**DOI:** 10.3390/diagnostics13081385

**Published:** 2023-04-10

**Authors:** Saravanan Srinivasan, Subathra Gunasekaran, Sandeep Kumar Mathivanan, Prabhu Jayagopal, Muhammad Attique Khan, Areej Alasiry, Mehrez Marzougui, Anum Masood

**Affiliations:** 1Department of Computer Science and Engineering, Vel Tech Rangarajan Dr.Sagunthala R&D Institute of Science and Technology, Chennai 600062, India; saravananjask@gmail.com; 2Department of Computer Science and Engineering, Sathyabama Institute of Science and Technology, Chennai 600119, India; subi10gks@gmail.com; 3School of Information Technology and Engineering, Vellore Institute of Technology, Vellore 632014, India; sandeepkumarm322@gmail.com (S.K.M.); j.prabhu@vit.ac.in (P.J.); 4Department of Computer Science, HITEC University, Taxila 47080, Pakistan; attique.khan@ieee.org; 5College of Computer Science, King Khalid University, Abha 61413, Saudi Arabia; areej.alasiry@kku.edu.sa (A.A.); mhrez@kku.edu.sa (M.M.); 6Electronics and Micro-Electronics Laboratory, Faculty of Sciences, University of Monastir, Monastir 5000, Tunisia; 7Department of Circulation and Medical Imaging, Faculty of Medicine and Health Sciences, Norwegian University of Science and Technology (NTNU), 7034 Trondheim, Norway

**Keywords:** rheumatoid arthritis, digital X-radiation image, consensus-based decision, faster-CNN, joint space narrowing, enhanced-region proposal network, artificial intelligence (AI)

## Abstract

We developed a framework to detect and grade knee RA using digital X-radiation images and used it to demonstrate the ability of deep learning approaches to detect knee RA using a consensus-based decision (CBD) grading system. The study aimed to evaluate the efficiency with which a deep learning approach based on artificial intelligence (AI) can find and determine the severity of knee RA in digital X-radiation images. The study comprised people over 50 years with RA symptoms, such as knee joint pain, stiffness, crepitus, and functional impairments. The digitized X-radiation images of the people were obtained from the BioGPS database repository. We used 3172 digital X-radiation images of the knee joint from an anterior–posterior perspective. The trained Faster-CRNN architecture was used to identify the knee joint space narrowing (JSN) area in digital X-radiation images and extract the features using ResNet-101 with domain adaptation. In addition, we employed another well-trained model (VGG16 with domain adaptation) for knee RA severity classification. Medical experts graded the X-radiation images of the knee joint using a consensus-based decision score. We trained the enhanced-region proposal network (ERPN) using this manually extracted knee area as the test dataset image. An X-radiation image was fed into the final model, and a consensus decision was used to grade the outcome. The presented model correctly identified the marginal knee JSN region with 98.97% of accuracy, with a total knee RA intensity classification accuracy of 99.10%, with a sensitivity of 97.3%, a specificity of 98.2%, a precision of 98.1%, and a dice score of 90.1% compared with other conventional models.

## 1. Introduction

Cancer starts when healthy cells in the breast change and grow out of control, forming a mass or sheet of cells called a tumor. A tumor may be benign or cancerous. Malignant means a cancerous tumor can grow and spread to other body parts. If a tumor is benign, it can enlarge but not spread. Radiologists must use a computer-aided detection (CAD) system to differentiate between normal and cancerous cell growth [1]. Arthritis is a form of joint inflammation that causes pain, stiffness, swelling, and limited movement. In India, 15% of the adult population had arthritis in 2015, compared with 22.7% in the United States [2]. According to medical research, there are around 100 different types of arthritis, with rheumatoid arthritis (RA) and osteoarthritis (OA) being the most frequent [3]. Medical imaging is a growing field and has several tools and methods for getting information from medical images. Magnetic resonance imaging (MRI) is one of the most extensively employed imaging techniques. MRI image samples are obtained using Brainix and neuroimaging data sources [4].

Up to 17% of people worldwide have progressive RA, an inflammatory condition. Most of the time, the condition causes inflammation in the joints, which makes them hurt for life, shortens people’s lives, and makes life less enjoyable. Some of the first signs of arthritis are narrowing joint space, breaking cartilage, less synovial fluid, and a torn meniscus. In both men and women, although more typically in middle-aged women, it can also affect organs and systems such as the skin, heart, and lungs [5]. RA is a long-lasting inflammatory disease that painfully destroys joints. Still, measuring how well a therapy works greatly depends on how the patient and the clinician feel about it. Aggressive disease-modifying therapy can alleviate symptoms and avoid irreversible joint deterioration [6]. Patients with RA who are getting good medical care may still develop musculoskeletal problems, such as damage to the knee joints that cannot be fixed. Total knee arthroplasty (TKA) surgeries on RA patients may become more common and have more complications as patients age. Pathogenesis, prognosis, and medical treatment for RA of the knee differ from those for OA of the knee, so the expected results of TKA are also different. The advanced knee damage RA causes is not a local problem, but a sign of a systemic problem. These variables influence the outcomes of TKA [7]. Research into the ways that artificial intelligence (AI), especially machine learning (ML) and deep learning (DL) techniques, could be used in medicine and healthcare is growing quickly. When there is not a good treatment for people with chronic rheumatological disorders such as RA, the second most common autoimmunity, these strategies can be very important for giving good care [8]. RA, an autoimmune disorder, affects between 0.5 and 1 percent of adults globally. The condition is most prevalent between the ages of 50 and 60, and women are twice as likely as men to develop it. Despite the lack of knowledge of what brings on rheumatoid arthritis, the consequences of this condition can be quite severe [9]. 

In 2010, the European League Against Rheumatism (EULAR) made its first suggestions about how to treat RA with biological and synthetic disease-modifying antirheumatic drugs (DMARDs). They gave rheumatologists, patients, payers, and other interested parties a summary of what is known now and what European specialists think based on evidence. Targeted synthetic and biologic DMARDs have transformed the treatment of this disorder. However, there are concerns about applying these novel ideas to clinical practice [10]. People with RA can now stop irreversible joint damage and, in some cases, heal damaged joints with the help of new drugs, biologics, and intensive treatment plans. Radiographs may reveal subchondral bone sclerosis and osteophyte formation as injured joints heal. The “evaluation of RA by scoring big joint destruction and healing in radiographic imaging” (ARASHI score) says that these changes, called secondary osteoarthritis (OA), are signs of rheumatoid arthritis. Radiographs include them in the definition of structural remodeling of large joints and recommend them as a way to make large joints more stable [11]. Scientists have used the term “human activity recognition” (HAR) to describe how machines automatically analyze what people do. Healthcare professionals rely heavily on this to promote health and wellness. For example, aged monitoring, exercise supervision, and rehabilitation monitoring all use it. Human activity recognition (HAR) can be used to track and analyze the activity levels of the elderly so that we can detect health issues prematurely [12]. This semi-supervised hierarchical convolutional neural network (SS-HCNN) aims to get around how hard it is to put images into categories and how the annotations limit what can be done. The concept is to use unsupervised clustering of the low-level characteristics to divide images into a tree-like structure and then train a tree-like network of convolutional neural networks (CNNs) at the root and parent nodes using the generated cluster labels [13]. This study proposes a probability-based real-parameter encoding operator. The approach also reduces the chromosomal length, saving computation space. The proposed revised GA algorithm passed the two-step validation process. First, a typical DFJSP demonstrated the algorithm’s utility. Second, the algorithm solved a real-world problem. A Taiwanese fastener factory applied historical data to 100 and 200 work orders. The proposed GA method change solved GA encoding problems. They completed the 100 sets of work orders in 102 days and 39 min using the proposed GA, saving 3878 min (150,797 min – 146,919 min = 3878 min) [14]. The author used the 3D articular bone shapes of the hand joints of people with RA and psoriatic arthritis and healthy controls to train a new neural network.

High-resolution peripheral computed tomography (HR-pQCT) data from the head of the second metacarpal bone were used to make the bone. Using GradCAM, it generated heat maps of problem areas. After training, we gave the neural network patterns of arthritis that could not be categorized as RA, PsA, or HC to figure out what kind of arthritis they were. In 932 HR-pQCT images of 617 patients, hand bone shapes were obtained. The network discriminated HC, RA, and PsA with area-under-receiver-operator curves of 82%, 75%, and 68%, respectively. Heat maps showed bare spots and places where ligaments connect that are likely to erode and form bone spurs. Based on joint form, the neural network identified 86% of UA data as “RA,” 11% as “PsA,” and 3% as “HC” [15]. The author came up with a new, simple, and computer-generated method for diagnosing knee RA that is based on deep CNNs and automatically measures the severity of RA in knee joints. This network was trained using around 1650 digital X-radiation images of the knee joint from the Optical Knee X-radiation Images Mendeley Collection. We performed the validation procedure for 20% of the data [16]. 

The proposed method was based on spatial analysis, and they separated the edges of the skin using a method based on intensity. The thresholding algorithm for segmenting bone regions, the hit-or-miss transformation for segmenting bone lines, and a distance measure for detecting the joint region following localization was followed. The synovial region was then identified using the active contour technique. In arthritic situations, synovitis also develops. We divided this condition into four types based on how much fluid builds up in the synovial area. The various grades were defined and analyzed using deep learning. A deep learning system used a convolutional neural network to determine the exact grade of synovitis and the type of arthritis. The validation produced an average true-positive percentage of 88.52%, ranging from 78.12% to 98.95%. False positives fell by 1.41 percent. These findings demonstrate that the network efficiently differentiates synovial grades [17]. In a big data set, the machine-learning-based ensemble analytical approach (MLEAA) has two parts: learning and predicting. During the learning phase, Hadoop’s map-reduce technology processes the data, while the highlighted attributes move the prediction stage forward. Three different algorithms, including Ababoost, SVM, and ANN, were used in the proposed MLEAA approach’s prediction phase, and the final predictive value was computed based on the voting system [18]. There are ways to conduct traditional statistical modeling, but they limit these methods in how much data they can effectively analyze. It is necessary to create thorough, patient-specific prediction models. Techniques such as data mining and machine learning should be used to help make these kinds of models. Although it will be difficult, current technology should allow for the sub-grouping of patients with OA, which could improve clinical judgment and advance precision medicine [19]. Early knee osteoarthritis detection is presented. Deep-learning-based feature descriptors on X-radiation pictures perform this. Training and testing use the Mendeley Dataset VI. Deep learning-based feature descriptors on X-radiation pictures perform this CNN with LBP and HOG used joint space width to obtain the proposed model feature from the region of interest. KOA was classified using KL, SVM, RF, and KNN. Images were five-fold validated and cross-validated. The method achieved 97.14% cross-validation and 98% five-fold validation accuracy. In the future, it will be possible to combine the proposed method with other models to find diseases above and below the knee more complexly. Feature fusion can detect and classify different diseases [20]. The model contained both a joint-detection step and a joint-evaluation step. There were 216 radiographs taken from 108 RA patients, with 186 assigned to the training/validation dataset and 30 to the test dataset. In the training/validation dataset, pictures of the PIP joints, the thumb’s IP joint, and the MCP joints were manually cropped, evaluated by clinicians for joint space narrowing (JSN), and then augmented. To train and test a deep convolutional neural network for joint evaluation, 11,160 images were used. The joint detection machine learning system was trained using 3720 carefully selected images. Putting these two methods together made a model for figuring out how badly radiographic finger joints damage it. With a sensitivity of 95.3%, the model estimated JSN and erosion for the PIP, thumb IP, and MCP joints. JSN’s accuracy range was 49.6–65.4 percent, while its erosion range was 70–74.1 percent. The correlation between model and clinician scores per image was 0.72 to 0.88 for JSN and 0.54 to 0.56 for erosion [21]. 

The proposed system architecture was made up of a CNN layer and a multilayer-based metadata learning layer. This was conducted so that the information was reliable. Sparse coding estimates and metadata-based vector encoding were used for the additional dimension. To keep the geometric format of supervised data [22], a well-structured k-neighbored network was used to build nearby limitation atoms. This study proposes SVM-based detection of finger joints and mTS score estimation. Using X-radiations of 45 RA patients, the suggested approach recognized finger joints with 81.4% accuracy and evaluated erosion and the JSN score with 50.9% and 64.4% accuracy, respectively [23]. The proposed model scored JSN and erosion for PIP, thumb IP, and MCP joints with 95.3% sensitivity. JSN had an accuracy range of 49.6–65.4% and an erosion range of 70–74.1%. They correlated the model and clinician scores per image at 0.72–0.88 for JSN and 0.54–0.75 for erosion [24]. The accuracy of the modified pre-trained GoogleNet model was 89%, whereas that of the proposed custom model was 95%. Google Net had a sensitivity of 84% and a specificity of 90%. Model number three was 95% sensitive and 94% specific. When extracted features from customized models (SIFT + CNN) are compared with those from ML classifiers, the custom3 model performed better [25]. The suggested method was compared with other fuzzy clustering methods that are already used to show how well it works. We compared the support vector machine (SVM), Decision Tree (DT), rough set data analysis (RSDA), and fuzzy-SVM classification algorithms to find the best way to group things [26]. The authors aim to create an AI-based computer-aided diagnosis tool that can classify abnormalities by reading chest X-radiations and help doctors make an accurate diagnosis quickly. We used a Google-created convolutional neural network (CNN) called XceptionNet to find those pathologies in the ChestX-radiation-14 data. Additionally, the same data are being used to run other CNN-ResNet algorithms [27]. At the 100th training iteration, the mean square error and the false recognition rate dropped below 1.1%, suggesting that the LPRNN was trained correctly. Edge preservation index values were above the experimental threshold of 0.48, signal-to-noise ratios (SNRs) were greater than 65 dB, peak SNR ratios were greater than 70 dB, and destruction times were faster [28]. Principal Component Analysis improved characteristics, while the Co-Active Adaptive Neuro-Fuzzy Expert System sorted images of the brain into glioma or non-glioma groups. The PCA and CANFES classification techniques had a sensitivity (Se) of 97.6%, a specificity of 98.56%, an accuracy of 98.73%, a precision of 98.85%, a false positive rate of 98.11%, and a false negative rate of 98.185 [29]. Table 1 illustrates the various state-of-the-art methods for knee RA classification.

### 1.1. Contribution

The proposed system predicts the minimal joint space narrow region and knee RA severity grade value.The proposed system’s experimental analysis was carried out using various criteria. The RA severity classification parameters such as sensitivity, specificity, accuracy, precision, and dice score.Our proposed system classification paradigm outcomes perform better than traditional techniques.

### 1.2. Organization of Work

Section 1: this paper also discusses the techniques for dataset validation and inflammatory mediator ground truth production and discusses the different state-of-the-art method performances. Section 2: following the data collection step, pre-processing and segmentation of the thermograms take place. Section 3: in the last step, the algorithm differentiates between abnormal and normal knee thermograms and then divides aberrant knee thermograms into three distinct categories. Section 4: provides the RA classification results from various parameters and compares the results of various existing techniques.

## 2. Materials and Methods

The primary goal of this study is to examine whether or not a deep learning strategy is effective for RA categorization. In our presented system, we use two approaches like; (i) feature extraction for ROI localization using a deep learning model (active F-CRNN + Hybrid ResNet101 with domain adaption); and (ii) feature selection via a supervised learning technique (marginal joint space narrowing region). To classify the severity of RA in the knee, AlexNet was used. The following procedures were carried out on our system.

### 2.1. Materials

The study encompassed patients with RA symptoms who were older than 55. (knee pain, rigidity, palpitation, and impaired functioning). The BioGPS database repository provided the digitized X-radiation images of the patients (805 men and 1207 women), which are a publicly accessible dataset; hence, the total number of patients is 2012. We discarded 181 radiographic images out of a total of 3353 records for reasons such as postoperative assessment, injury, and infection. Thus, only 3172 X-radiation images were acquired for analysis. Knee X-radiation digital images start in the DICOM format, but are easily converted to the universal JPG format for further use [30]. The digital X-radiation of the knee joint had a resolution of around 3000 by 1500 pixels. Before conducting the analysis, the image brightness was standardized. Three medical domain specialists reviewed each digital X-radiation set (from the Dindigul scan center, Dindigul). Medical domain experts manually examined each digital X-radiation image in order to obtain two ground truth data points (minimum joint space narrowing area and RA classification using CBD grading criteria). Table 2 displays the numerous consensus-based decision grades used in analyzing rheumatoid arthritis.

Table 3 depicts the total number of digitized knee X-radiation images and CBD grading evaluations by three clinical professionals. We used 80% of the data for training, and we further split the training data into training (70%) and validation (10%). The remaining data (20%) were used for testing. The split-up of the dataset is displayed in Table 4.

### 2.2. Methods

The initial step involved reducing an image to a size of 227 × 227 × 3. We used two convolutional neural networks for RA detection and classification in the second phase (Hybrid ResNet101 and VGG16). The image characteristics must be sufficient for accurate CBD grade determination and effective RA classification. We identified marginal joint space narrowing and categorized RA using these convolutional neural networks to extract instructive visual characteristics. To complete the RA classification process, two CNNs were used. First, we used ResNet101 with a domain adaptation strategy to identify marginal joint space contraction. Second, we used VGG16, which was trained using a domain adaptation technique, to classify RA. Finally, we evaluated the method’s effectiveness and contrasted our findings with those of other similar techniques already in use. Figure 1 shows a flowchart of the recommended process.
(1)$${Loss}_{local}\left(t_{i}^{a},b\right)=\sum_{u\in \{m,n,y,z\}}{smooth}_{f_{n}{Loss}_{local1}}(t_{i_{1}}^{a}-b_{i})$$
(2)$${smooth}_{f_{n}{Loss}_{local1}}\left(p\right)=\left\{\begin{array}{c} 0.5p^{2}if\left|p\right|<1\\ \left|p\right|-0.5\;otherwise, \end{array}\right.$$

The above Equations (1) and (2) represent the overall loss value (${Loss}_{local})$ of an enhanced region proposal network along with an association of the classification loss. (${smooth}_{f_{n}{Loss}_{local1}})$, ${Loss}_{local1}$ is robust loss, $b_{i}$ is aground-truth regression targets, and $t_{i_{1}}^{a}$ is a predicted tuple. p is computed by a softmax.

#### 2.2.1. Determination of Joint Space Narrowing

In deep learning, the F-CRNN is just one of the methods. The faster CRNN architecture is now the standard object identification method because of its ability to anticipate and score single or numerous items in an image. The enhanced-region proposal network and F-CRNN are two integral parts of the F-CRNN network. To ensure that the Quick R-CNN module receives only the best region suggestions, the ERPN generates them. To identify areas of interest in digital X-radiation pictures, we trained the ERPN. The end-to-end convolutional network (ERPN) can accurately anticipate the boundaries and scores of objects of interest at any coordinate. The ResNet101 network was used for F-CRNN feature extraction. 

Each convolutional layer in ResNet101 was followed by a batch normalization and activation layer (ReLU). By avoiding parallel connections to the typical layers, this architecture facilitated more efficient training of deep neural networks. Features were extracted, and convolutional feature maps were generated using a combination of convolutional and max-pooling layers. Image characteristics were fed into ERPN, and region suggestions were generated as outputs. The ROI pooling layer took the feature vectors from the function maps. Each vector function was linked to the underlying layer. We individually trained the ROI detection model for the AP view’s medial and lateral compartments. When the algorithm produced several ROI detections, we chose the ROI with the highest prediction accuracy for each knee joint. To evaluate the proposed model, we counted the narrow regions of the marginal joint space that achieved IoU ≥ 0.70. As a result of the detection, we saved the predicted bounding boxes. We used weights that had already been trained on ResNet-101, and then used the domain adaptation method to fine-tune them. Figure 2 shows how modified ResNet-101 can find approaches with a narrow joint space in the knee. The most important part of the Faster R-CNN architecture is ERPN. ERPN predicts the scores of objects and their locations. The algorithm compares the narrow areas of the knee joint space in the medial and lateral compartments to find the narrow area in the middle. The best thing about this method is that it can find even the smallest changes in knee joint space.

#### 2.2.2. RA Classification

For knee RA severity classification, we conducted this research using a modified version of the VGG16 architecture and a domain adaptation technique, as shown in Figure 3. The VGG16 model was made up of five convolutional layers, three max-pooling layers, and three fully connected layers—all the digital X-radiation images needed to be resized to (227 × 227 × 3). In our implementation, X-radiation image information for training purposes accounted for eighty percent of the total, while X-radiation image information for evaluating purposes accounted for twenty percent. Although there are sixteen layers in VGG16, only a subset of those layers is required for feature extraction. In order to shorten the amount of time needed for training and establish more control over the fitting process, we assigned a dropout ratio value of 0.5 to the completely connected layer (fcl6) and the fully connected layer (fcl7). The characteristics were taken from the fully connected layers designated fcl6 and fcl7, respectively. To categorize the retrieved features into 1000 categories, VGG16’s architecture used a fully connected layer (fcl8). Then, we conducted one last round of tuning on the pre-trained VGG16 model’s ability to classify RA by changing parameters in the model’s last three layers. The model’s last three layers were swapped out for a fully linked layer, a softmax layer, and a classification layer. In addition, a newly linked layer was assigned to five groups of RA grades for the dataset: Grade 0, Grade 1, Grade 2, Grade 3, and Grade 4. We trained the proposed network by using digital knee X-radiations, a small-batch test dataset, gradient descent, and maximal epochs. Our proposed network learning strategy used stochastic gradient descent, and we compared its performance to previous efforts. The proportion of knee X-radiation images from the test set for which the network correctly predicted the RA grade was used to calculate proposed work accuracy. The proposed approach achieved an overall accuracy of 99.10% in classifying knee RA cases. Table 5 illustrates the Visual Geometry Group (VGG16) CNN operation for RA grade classification. Figure 4, depicts the RA classification using VGG16 architecture.

## 3. Experimental Results and Discussion

### 3.1. Experimental Parameter Settings

For our investigation, we utilized a machine with 8 GB of RAM and 256 GB of SSD, an Intel i3-core CPU, and Radeon R2 graphics. For image processing, we selected MATLAB 2020-a. Each stochastic gradient descent iteration used a batch size of 256 for both the F-CRNN and the enhanced-region proposal networks, and a learning rate of 4e-3 was applied. The presented approach method corresponds to around 4 h of model training, and the maximum number of iterations was 0.6 k.

### 3.2. Detection of Marginal Knee Joint Space Narrowing

Sample images of the marginal joint space narrowing region of interest can be seen in Figure 5. The IoU (Intersection of Union) metrics were used to evaluate our region of interest detection system. This metric was the size of the intersection between the area of the actual bounding box and the area of the predicted bounding box divided by the size of the area of both boxes added together. When the IoU was 0.70, the narrow marginal joint space was found in 99.72% of the knee joints using our presented model. Additionally, Figure 6 depicts the ROC curve for marginal joint space narrow detection. The results of the presented marginal joint space narrow detection model obtained a sensitivity rate of 98.67%, a Dice score of 98.58%, a precision rate of 98.46%, a specificity rate of 98.50%, a false positive rate of 0.0100, a false negative rate of 0.0197, and an overall accuracy rate of 98.97%, as shown in Table 6, and the graphical illustration of Table 6 values is depicted in Figure 7. Table 7 demonstrates the metric performance outcomes of the proposed ResNet101 and VGG16 model to classify the RA. From Table 7, the outcome of the VGG16 outperforms the well-pre-trained ResNet101 model in classifying RA.

### 3.3. Parameter Metrics for Performance Computation

In our presented systems, classification accuracy analysis was computed by five different performance metrics: sensitivity, specificity, precision, accuracy, and Dice score.
(3)$$Sensitivity=\frac{\beta }{\beta +ø}$$
(4)$$Specificity=\frac{ø}{ø+\beta }$$
(5)$$Precision=\frac{\beta }{\beta +µ}$$
(6)$$Accuracy=\frac{\beta +ø}{\beta +µ+nø+\gamma }$$
(7)$$Dicescore=\frac{2*\beta }{\left(\beta +\gamma \right)+(\beta +µ)}$$

Here, β-represents the true positive values, ø-represents the true negative values, µ-represents the false negative values, and γ-represents the false positive values.

### 3.4. Intensity Classification of Rheumatoid Arthritis

The presented model achieved 99.10% accuracy on the whole test set. The confusion matrix of the presented method is shown in Figure 8, and its performance is compared in detail to that of current methods in Table 5. In Figure 8, we examine the training and learning procedure as a whole to assess the planned activity’s success. Table 6 demonstrates the highest accuracy rate for classifying CBD grades zero–three–four knee joints. The knee joints with a CBD grade of one or two are the toughest to categorize. As can be seen in Figure 8, there is only a marginal amount of room for error when classifying knee joints as CBD Grades zero, three, or four. Knee joints that are classified as CBD Grades one or two have a small number of marginal misclassifications. In several circumstances, the proposed approach incorrectly estimated CBD Grade two as Grade one and vice versa. Joint space narrowing and bony spur development are significantly different in CBD Grade four knee joints. However, CBD-grade one knee joints show little change in JSN or osteophyte growth compared with the other classes. Types of knee RA and their intensity levels are shown in Figure 5. Table 8 and Figure 9 illustrate the JSN accuracy of the proposed and other state-of-the-art methods comparison.

In this study, we developed a deep learning model to automatically grade the severity of knee RA using a consensus-based approach. We compared the proposed work to prior strategies and found that it outperformed the competition. At the elementary level, notably in Grade one and Grade two, we found that our method differed from that of the medical professionals. We evaluated the presented work by comparing its results with similar existing studies. Compared with previously existing models, the presented work (a knee joint space narrowing diagnosis and class label) fares very well. It takes about 7 h of training to reach 0.6 k iterations. The outcomes of the presented methodology are shown in Table 9, which includes the outcomes of each CBD grade individually. Multiple metrics were employed to estimate the model’s performance, as indicated in Table 10. Figure 10 and Figure 11 depict the ROI curve for RA severity classification for both knees.

Figure 12 shows that the presented system outperformed other methods in terms of sensitivity (Se), specificity (Sp), precision (Pr), accuracy (Acc), and dice score (Ds), demonstrating deep learning’s capability. Figure 13 depicts the CBD grade outcome doughnut chart. In this research, the presented model increased overall ROI detection accuracy by up to 0.5 percent and improved classification accuracy by up to 1.18 percent. The proposed model is more dependable as a result of the detailed knee JSN characteristics. The improvement was satisfactory, and we agree with the observation that the AP view has a significant portion of the information necessary to assess the severity of knee RA with the CBD grading system. The CBD score is often examined using the AP view alone. Table 6 presents a comparison of the output of the proposed methodology with that of other methods that are currently in use. R K Ahalya et al. (2022) obtained Se of 0.9491, Sp of 0.9408 Pr of 0.9213, Acc of 0.9551 and Ds 0.8991; Uma Ramasamy et al. (2022) achieved Se of 0.8891, Sp of 0.8982, Pr of 0.8844, Acc of 0.9012, and Ds of 0.8923; Shawli Bardhan et al. (2021) obtained Se of 0.9785, Sp of 0.9561, Pr of 0.9713, Acc of 0.9864, and Ds of 0.9231; Rabbia Mahum et al. (2021) achieved Se 0.9815, Sp of 0.9789, Pr of 0.9896, Acc of 0.9714, and Ds of 0.9795; Kristine et al. (2022) obtained Se of 0.9012, Sp of 0.9101, Pr of 0.8915, Acc of 0.9211, and Ds of 0.9117; Shawli Bardhan et al. (2020) achieved Se of 0.8889, Sp of 0.8541, Pr of 0.8114, Acc of 0.8671, and Ds of 0.8781; and Sujeet More et al. (2022) obtained Se of 0.9622, Sp of 0.9771, Pr of 0.9831, Acc of 0.9685, and Ds of 0.9121. Our active deep CNN model acquired a knee joint identification accuracy of 98.97% and a knee RA severity classification accuracy of 99.10% using the presented methodology. This model also gives superior performance to handmade features. The active deep CNN model that we have presented and the pre-trained domain adaptation models that are employed in our system produce improved prediction accuracy outcomes for the five classes of knee RA that were experimentally determined.

## 4. Conclusions and Future Work

In this study, we propose a way to find and classify rheumatoid knee arthritis by using a deep convolutional neural network (CNN). We use the domain adaptation strategy to use already-trained models such as ResNet101 and VGG16. We evaluate the results against standard methods. The results of our experiments show that our proposed method is better at diagnosing rheumatoid knee arthritis than the current best practices. The presented approach achieved an classification accuracy of 98.97% and 99.10%. We used active deep CNN to predict and grade knee RA and then compared our results to work performed in MATLAB 2020a before. In this study, we provide a deep learning method for the automated detection and classification of RA in the knee. The presented methodology would analyze digital X-radiation pictures of the knee to identify the ROI (minimum knee joint space narrow area) and determine the degree of rheumatoid arthritis. Soon, we want to use this method to assign grades to MRI scans of knees affected by rheumatoid arthritis. 

A potential direction for future research might be developing a system to assist medical professionals in identifying the location and cause of knee inflammation using thermogram images as a secondary diagnostic tool. The size of the dataset will also be increased so that temperature-flow patterns specific to arthritis can be made for better classification. Additionally, the presented method can be used with other models to find diseases other than knee problems in a hybrid and flexible way. Even more, this may be combined with feature fusion techniques for diagnosing and categorizing a wide range of additional disorders.

## Figures and Tables

**Figure 1 diagnostics-13-01385-f001:**
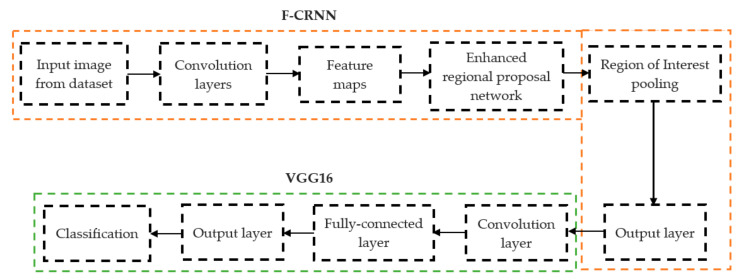
Presented association of F-CRNN and VGG16 layered architecture.

**Figure 2 diagnostics-13-01385-f002:**
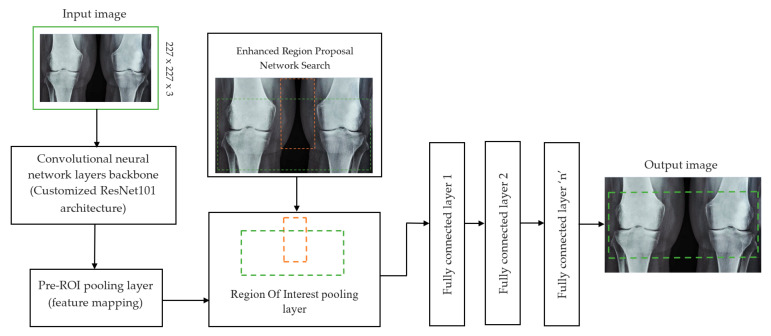
Association features of F-CRNN and cutomized ResNet101 architecture.

**Figure 3 diagnostics-13-01385-f003:**
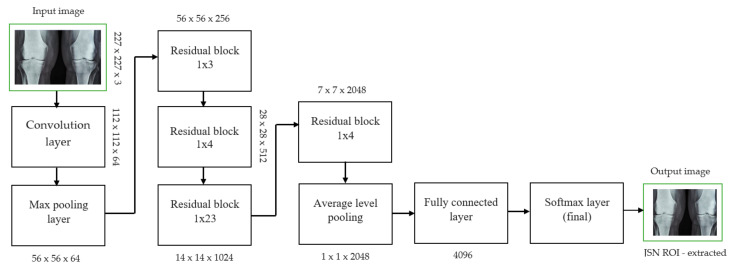
JSN ROI extracted for knee joint narrow space from ResNet101 architecture.

**Figure 4 diagnostics-13-01385-f004:**
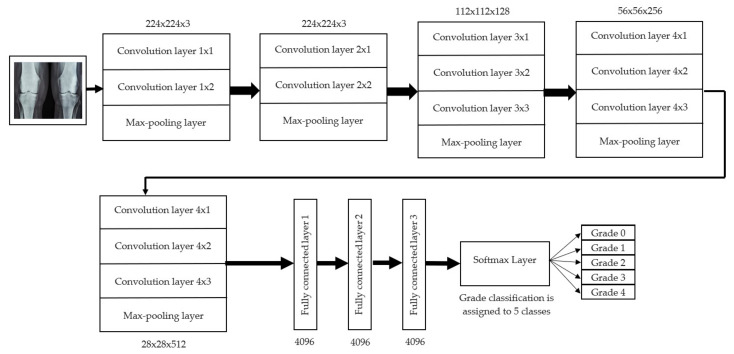
RA classification using VGG16 architecture.

**Figure 5 diagnostics-13-01385-f005:**
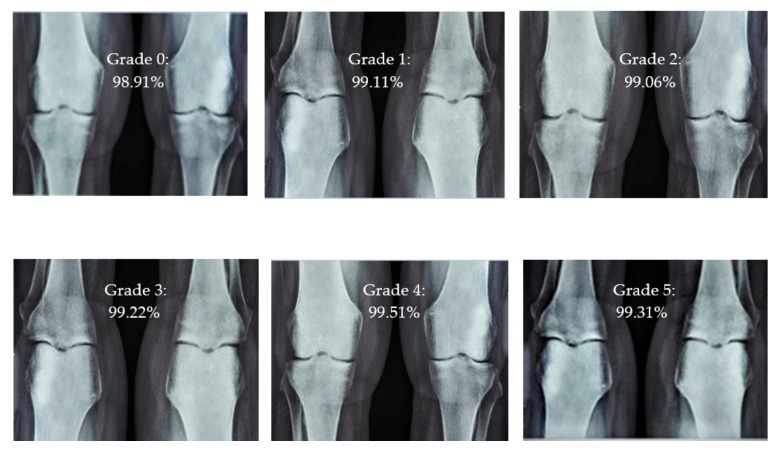
Different CBD grade levels of knee rheumatoid arthritis.

**Figure 6 diagnostics-13-01385-f006:**
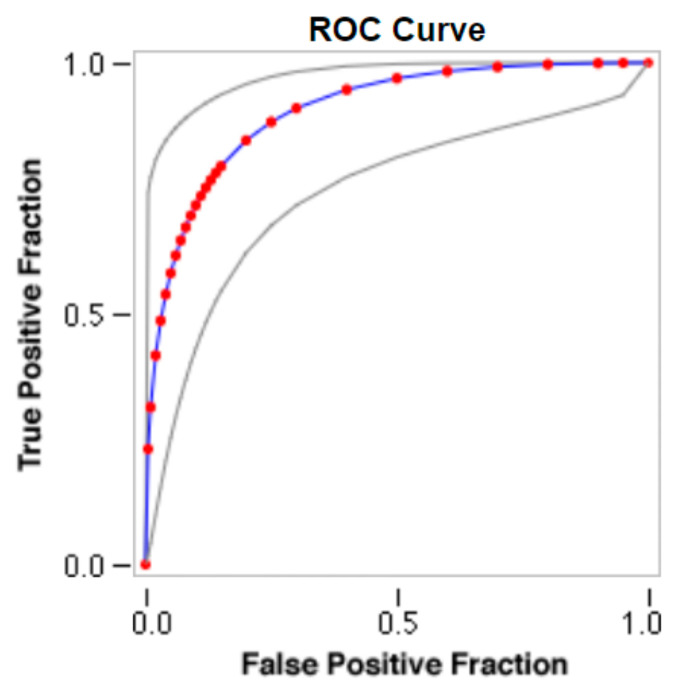
ROC curve for marginal joint space narrow detection.

**Figure 7 diagnostics-13-01385-f007:**
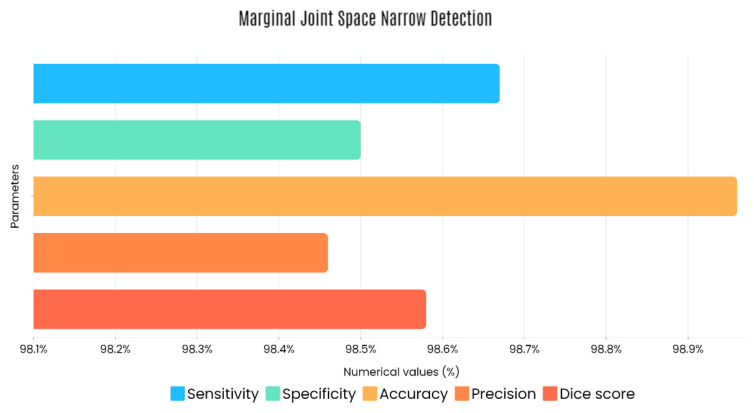
Graphical illustration of obtained marginal JSN detection parameters.

**Figure 8 diagnostics-13-01385-f008:**
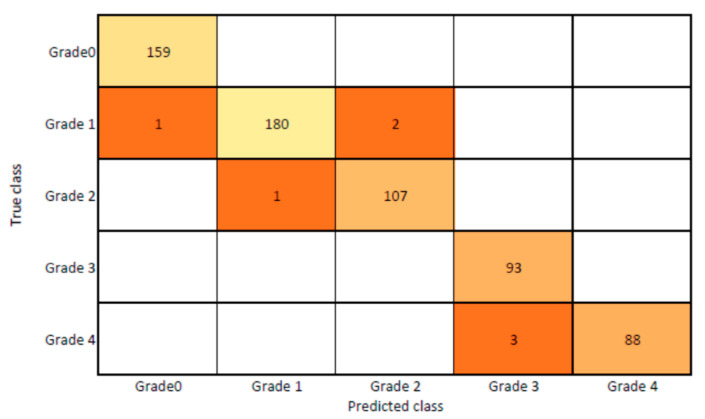
Proposed system confusion matrix (manual vs. automatic) for grade classification.

**Figure 9 diagnostics-13-01385-f009:**
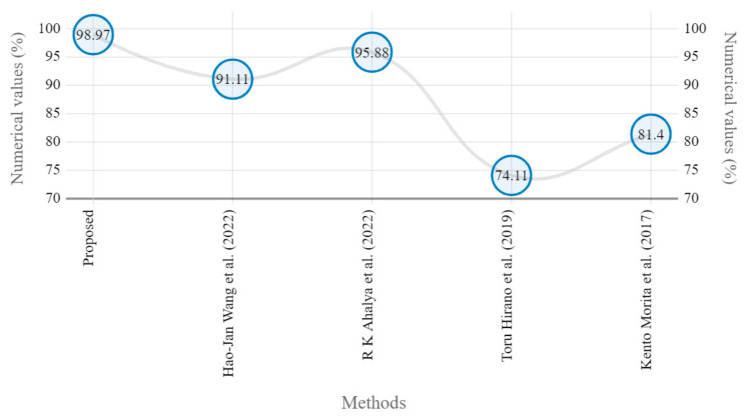
Marginal JSN accuracy comparison of proposed and state-of-the art methods [5,22,24,25].

**Figure 10 diagnostics-13-01385-f010:**
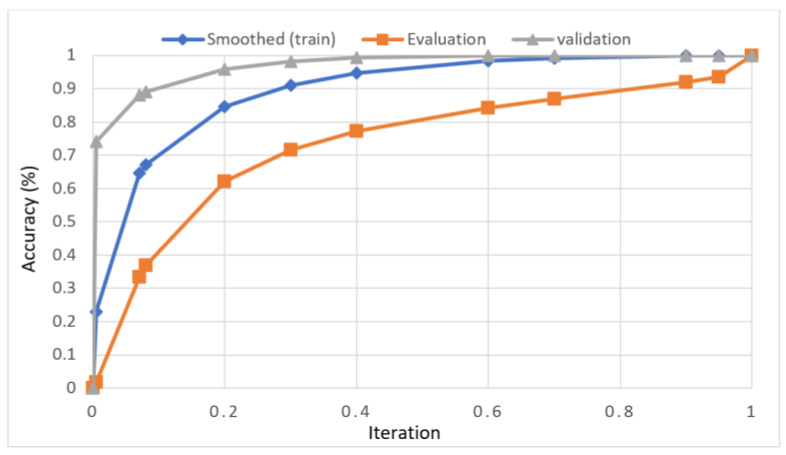
ROI curve for RA severity classification for knee-1.

**Figure 11 diagnostics-13-01385-f011:**
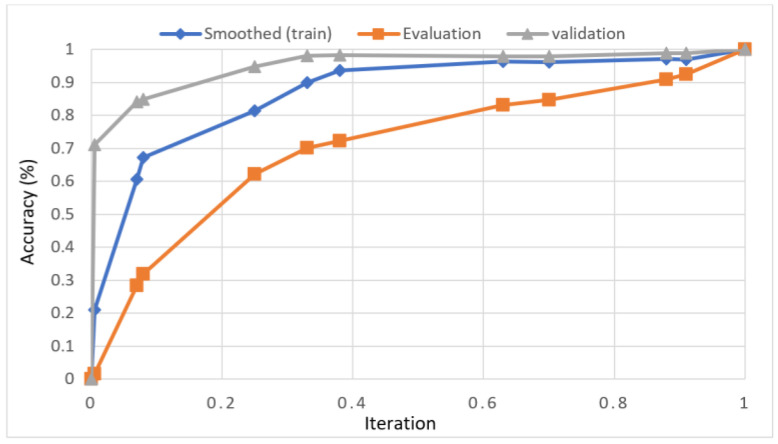
ROI curve for RA severity classification for knee-2.

**Figure 12 diagnostics-13-01385-f012:**
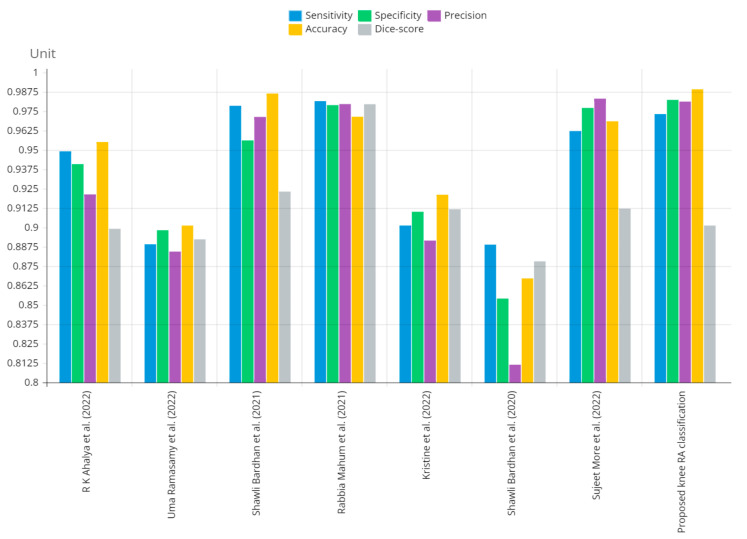
Graphical annotation of presented and conventional methods metric comparison.

**Figure 13 diagnostics-13-01385-f013:**
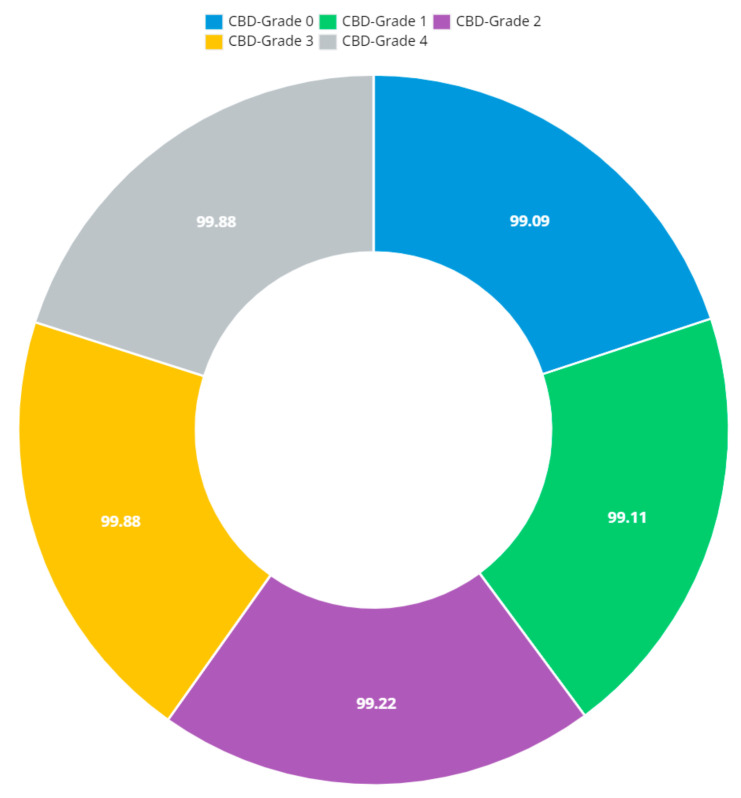
Doughnut chart of CBD grade outcomes.

**Table 1 diagnostics-13-01385-t001:** Knee RA Grade Classification from different state-of-the-art methods.

Author	Year	Dataset	Technique	Outcomes	Merits
Sharon Ho [19]	2018	Credible	Bagging, J48, and AdaBoost	Obtained classification accuracy was 96.21%	For the RA dataset, bagging is a better way to classify than the other options.
Lukas Folle [20]	2022	High-resolution peripheral-computed-tomography	Novel neural network on 3D	Obtained RA classification accuracy was 86%	The trained network could group nonspecific disorders such as UA based on joint form.
Apoorva Parashar [21]	2022	Mendeley data	Computer-generated diagnostic approach, based on the Deep CNN	Proposed model achieved 95.62% of accuracy in RA classification.	Automated RA intensity rating for knee joints.
RJ Hemalatha [22]	2019	MEDUSA	Spatial analysis using intensity-based approach	Obtained RA classification accuracy was 98.4%	Deep learning is used to define and analyze various grade levels and to determine the type of arthritis automatically.
S.Shanmugam [23]	2018	Built-in streaming	Machine-learning-based ensemble analytic approach	Proposed model achieved 85% of accuracy in RA classification	The final prediction value is computed using three distinct algorithms, including Ababoost, SVM, and ANN, and a voting mechanism, depending on the results.
Afshin Jamshidi [24]	2019	Osteoarthritis initiative	Data mining and machine learning	The model obtained 75% accuracy in predicting which patients would progress to having symptomatic OA	Comprehensive patient-specific prediction model
Rabbia Mahum [25]	2021	Mendeley	Knee osteoarthritis detection at early stages using deep-learning-based feature extraction and classification	The HOG features descriptor provides approximately 97% accuracy for the early detection and classification of KOA for all four grades of KL	About 97% of the time, a description of KOA based on HOG features is accurate for all four levels of KL severity.
Tengfei Yang [26]	2020	Metacarpophalangeal arthritis	Geometric and textural features of synovium thickening and bone erosion	Proposed model achieved 92.50% of accuracy in RA classification	Without the use of medical expert analysis or blood sample analysis (for example, detecting C-reactive protein, measuring erythrocyte sedimentation rate, and testing rheumatoid factor), this method indicates a substantial grade of metacarpophalangeal RA ultrasound imaging.
Toru Hirano [27]	2019	MEDUSA	Deep learning model to assess radiographic finger joint destruction in RA	Obtained RA classification accuracy was 95.3%	Together, they form the radiographic assessment model for finger joint damage.

**Table 2 diagnostics-13-01385-t002:** Consensus-based decision gradings.

Total No. of Images	CBD Grades (G)	RA Analysis
3100	G0	Benign
	G1	RA uncertainty
	G2	Initial stage of RA
	G3	Mitigate-observed stage of RA
	G4	Catastrophic

**Table 3 diagnostics-13-01385-t003:** Consensus-based decision from three experienced clinicians.

CBD Grades (G)	Severity	Clinician-1	Clinician-2	Clinician-3	Toal Agreed Images (Grade)
G0	Normal	774	802	769	2345
G1	Doubtful	821	850	811	2482
G2	Mild	610	598	621	1829
G3	Moderate	551	501	551	1603
G4	Severe	416	421	420	1257

**Table 4 diagnostics-13-01385-t004:** Training and testing phase analysis of a dataset.

Categorization	Training Phase (70%)	Evaluation Phase (10%)	Testing Phase (20%)
Total number of patients	1408	201	402
Total number of knees	2221	317	634

**Table 5 diagnostics-13-01385-t005:** VGG16 architecture for RA grade classification.

Layer		Feature Map	Size	Kernel Size	Stride	Activation Layer
Input	Image	1	224 × 224 × 3	-	-	-
1	2 × convolution	64	224 × 224 × 64	1 × 1	1	Relu
	Max-pooling	64	112 × 112 × 64	3 × 3	2	Relu
3	2 × convolution	128	112 × 112 × 128	1 × 1	1	Relu
	Max-pooling	128	96 × 96 × 128	3 × 3	2	Relu
5	2 × convolution	256	56 × 56 × 256	3 × 3	2	Relu
	Max-pooling	256	28 × 28 × 256	3 × 3	2	Relu
7	3 × convolution	512	28 × 28 × 256	1 × 1	1	Relu
	Max-pooling	512	14 × 14 × 512	3 × 3	2	Relu
10	3 × convolution	512	14 × 14 × 512	1 × 1	1	Relu
	Max-pooling	512	7 × 7 × 512	3 × 3	2	Relu
11	Fully connected	-	4096	-	-	Relu
14	Fully connected	-	4096	-	-	Relu
15	Fully connected	-	4096	-	-	Relu
16	Fully connected		1000	-	-	Softmax

**Table 6 diagnostics-13-01385-t006:** Marginal joint space narrow detection parameter outcomes.

Marginal JSN Detection	
Parameters	Obtained Values (%)
Sensitivity	98.67
Specificity	98.5
Accuracy	98.97
Precision	98.46
Dice score	98.58
False positive rate	0.01
False negative rate	0.0197

**Table 7 diagnostics-13-01385-t007:** Outcome comparison of ResNet101 and VGG16 models.

Models	Sensitivity (%)	Specificity (%)	Precision (%)	Accuracy (%)
ResNet-101 with DA	98.07	97.87	98.04	97.96
VGG16 with DA	98.78	98.67	98.71	98.81

**Table 8 diagnostics-13-01385-t008:** JSN outcomes of proposed and state-of-the-art methods.

Prediction of Marginal JSN Accuracy				
Models	Sensitivity (%)	Specificity	Precision	Accuracy (%)
Proposed model	98.3	97.98	98.46	98.97
Hao-Jan Wang et al. (2022) [25]	90.81	89.98	91.05	91.11
R K Ahalya et al. (2022) [5]	94.34	94.12	95.21	95.88
Toru Hirano et al. (2019) [22]	73.29	73.11	73.98	74.11
Kento Morita et al. (2017) [24]	80.9	79.98	80.97	81.4

**Table 9 diagnostics-13-01385-t009:** Presented and conventional methods performance metric comparison.

Techniques/Parameters	Measuring Parameters				
	Sensitivity	Specificity	Precision	Accuracy	Dice Score
Shawli Bardhan et al. (2021) [3]	0.9785	0.9561	0.9713	0.9864	0.9231
R K Ahalya et al. (2022) [5]	0.9491	0.9408	0.9213	0.9551	0.8991
Sujeet More et al. (2022) [6]	0.9622	0.9771	0.9831	0.9685	0.9121
Rabbia Mahum et al. (2021) [9]	0.9815	0.9789	0.9896	0.9714	0.9795
Kristine et al. (2022) [7]	0.9012	0.9101	0.8915	0.9211	0.9117
Uma Ramasamy et al. (2022) [31]	0.8891	0.8982	0.8844	0.9012	0.8923
Shawli Bardhan et al. (2019) [32]	0.8889	0.8541	0.8114	0.8671	0.8781
Presented knee RA classification	0.9731	0.9823	0.9812	0.9910	0.9012

**Table 10 diagnostics-13-01385-t010:** Consensus-based decision grade outcomes.

Grade-CBD	Classification of Knee RA Associated with Domain Adaptation			
	Recall	Accuracy	F1-Score	Precision
CBD-Grade 0	99.09	99.13	99.18	98.95
CBD-Grade 1	99.11	99.16	99.25	98.98
CBD-Grade 2	99.22	99.39	99.34	99.21
CBD-Grade 3	99.88	99.59	99.51	99.41
CBD-Grade 4	99.88	99.62	99.57	99.44

## Data Availability

Not applicable.

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
