# Peer review of "A Framework of Faster CRNN and VGG16-Enhanced Region Proposal Network for Detection and Grade Classification of Knee RA"

_diagnostics, 2023, doi:10.3390/diagnostics13081385_

Round 1

Reviewer 1 Report

A framework of faster CRNN and VGG16-Enhanced region proposal network for detection and grade classification of knee rheumatoid arthritis is proposed. It is interesting but not complete enough for publication. Some specific comments are listed as follows:

(1)    Some spelling mistakes and writing style can be further improved, these small mistakes need to be corrected before the paper is published.

(2)    Some of the statements are not formal enough to qualify as an academic paper.

(3)    The current abstract is too redundant to make people understand the novelty and significance of the paper quickly. Therefore, the abstract needs to be concise so that the reader can understand innovation and value of this paper more easily.

(4)    The format of Table 1 needs to be modified according to the requirements given.

(5)    Adjust the icon position to avoid unnecessary white space.

(6)    The formula descriptions of performance metrics are not conducive to reading, and symbols can be used instead of verbal descriptions.

(7)    The contents of some graphs and tables are duplicated, which is unnecessary.

(8)    Some images are not easy to read, for example, the legend is too small compared to the image.

(9)    In the introduction section, the author omitted some important references about classification methods based on deep learning, such as

https://www.doi.org/10.1109/10.1109/TIP.2018.2886758

https://www.doi.org/10.1109/TGRS.2023.3248040

https://www.doi.org/10.1109/JBHI.2019.2909688

Author Response

The response sheet is attached. 

Author Response

The response sheet is attached. thank you

Round 2

Reviewer 1 Report

I have no other questions.

Author Response

Thank you for your value able recommendation.